# Recent Advances in Synthesis of Non-Alternating Polyketone Generated by Copolymerization of Carbon Monoxide and Ethylene

**DOI:** 10.3390/ijms25021348

**Published:** 2024-01-22

**Authors:** Xieyi Xiao, Handou Zheng, Heng Gao, Zhaocong Cheng, Chunyu Feng, Jiahao Yang, Haiyang Gao

**Affiliations:** School of Materials Science and Engineering, PCFM Lab, GD HPPC Lab, Sun Yat-sen University, Guangzhou 510275, China; xiaoxy53@mail2.sysu.edu.cn (X.X.); zhenghdou@mail.sysu.edu.cn (H.Z.); gaoheng16@163.com (H.G.); chengzhc@mail2.sysu.edu.cn (Z.C.); fengchy28@mail2.sysu.edu.cn (C.F.); yangjh73@mail2.sysu.edu.cn (J.Y.)

**Keywords:** polyketone, carbon monoxide, ethylene copolymerization, degradation

## Abstract

The copolymers of carbon monoxide (CO) and ethylene, namely aliphatic polyketones (PKs), have attracted considerable attention due to their unique property and degradation. Based on the arrangement of the ethylene and carbonyl groups in the polymer chain, PKs can be divided into perfect alternating and non-perfect alternating copolymers. Perfect alternating PKs have been previously reviewed, we herein focus on recent advances in the synthesis of PKs without a perfect alternating structure including non-perfect alternating PKs and PE with in-chain ketones. The chain structure of PKs, catalytic copolymerization mechanism, and non-alternating polymerization catalysts including phosphine–sulfonate Pd, diphosphazane monoxide (PNPO) Pd/Ni, and phosphinophenolate Ni catalysts are comprehensively summarized. This review aims to enlighten the design of ethylene/CO non-alternating polymerization catalysts for the development of new polyketone materials.

## 1. Introduction

Polyketones (PKs) are a kind of polymer containing the carbonyl groups in the main chain, which are usually made by the copolymerization of carbon monoxide (CO) and olefins [1,2,3]. According to the olefin comonomer type, PKs are divided into aliphatic (aliphatic alkene) [1,4,5], aromatic (vinyl arene) [6,7,8,9,10,11,12], and cyclic PKs (cycloolefin) [13,14,15,16]. Among these three kinds of PKs, aliphatic polymers generated by the copolymerization of olefins and CO have attracted considerable attention over the last decades, especially as a cheap and plentiful ethylene monomer [17,18]. In this review, we focus on aliphatic PKs generated from the copolymerization of ethylene and CO, and aliphatic PKs are called PKs in the following text.

As a kind of engineering plastic, PKs show unusual properties such as conspicuous chemical resistance, impermeability to hydrocarbons, high impact strength, and good thermostability [17,18]. Various modifications [19,20,21,22,23,24,25,26,27] and post-functionalizations of PKs widely expand their potential applications in biomedicine, electronic devices, fluorescent materials, and textile fibers [17,28,29,30,31]. Most importantly, PKs are a kind of degradable polymeric material, and their degradation can be induced by UV light irradiation through the well-known Norrish I and II processes [32,33,34,35,36,37]. Undoubtedly, PKs are highly attractive polymeric materials for increasing white plastic pollution. On the one hand, the synthesis of PKs needs to consume polluting gas CO stock. On the other hand, degradable PKs are environmentally friendly polymeric materials.

The first example of the metal-catalyzed synthesis of PKs was reported in 1951, in which CO/ethylene copolymerization was catalyzed by K_2_Ni[CN]_4_ to produce low-melting oligomers [38]. In the subsequent decades, various nickel and palladium catalysts have been developed for alternating copolymerizing ethylene and CO to synthesize aliphatic PKs [39,40,41,42,43,44,45,46,47]. A breakthrough is the diphosphine palladium catalysts discovered by Drent in 1991, which are highly efficient for alkene/CO copolymerization [48]. In 1996, a CO/ethylene/propylene terpolymer called “Carilon” was successfully commercialized by Shell company, but production of “Carilon” was closed in 2000 due to strategic changes [2]. Hyosung company began to produce the same PKs in 2013, and the current annual production of PKs has reached 50,000 tons [49]. These engineering plastics are commercially available for food contact, industry, and medical equipment.

Because the insertion of CO into the five-membered Pd-acyl-chelates is kinetically favored over ethylene insertion and successive insertion of CO is thermodynamically prohibited, perfect alternating PKs are often produced by Ni/Pd catalyzed copolymerization of ethylene and CO. In the early period, perfect alternating PKs received considerable attention in both academic and industrial fields (Figure 1). Recently, PKs without a perfect alternating structure, including non-perfect alternating PKs and polyethylene (PE) with in-chain ketones, have also attracted increasing interest (Figure 1). In 2002, non-alternating CO/ethylene copolymerization was first realized using phosphine–sulfonate Pd catalysts developed by Drent [50]. With improved flexibility, non-alternating PKs have potential applications in many industrial fields such as textile fibers and packing films. PE with in-chain ketones was recently developed and exhibited similar mechanical properties to commercial high-density polyethylene (HDPE) and better degradability and hydrophily [51,52]. Introducing a small amount of CO in polyolefins provides a promising strategy to balance the demand for polyolefins and expectations of environmental friendliness, and PE with in-chain ketones can be understood as degradable HDPE.

Perfect alternating PKs have been systematically summarized in previous reviews [1,3,4,5,53,54,55,56,57]. In this review covering the literature published until the end of 2023, we focus on recent advances in the synthesis of PKs without a perfect alternating structure including non-perfect alternating PKs and PE with in-chain ketones. The chain structure of PKs without a perfect alternating structure, catalytic copolymerization mechanism, and Ni/Pd catalysts are summarized.

## 2. Chain Structure of PKs without a Perfect Alternating Structure

PKs without a perfect alternating structure have sequential ethylene units in a polymer chain, which originates from sequential ethylene insertion during the copolymerization of ethylene and CO. The resulting microstructure and melting temperatures (*T*_m_) of three kinds of PKs are shown in Table 1.

Based on the arrangement of the ethylene and carbonyl groups, the structural fragments and corresponding NMR shifts of PKs without a perfect alternating structure are classified in Table 2.

Carbonyl content and extra ethylene (C2) insertion are the two most important structural parameters of PKs, defined as total mol % CO units relative to all monomer units [59] and total mol % ethylene of copolymer involved in non-alternation [50], respectively. Carbonyl content can be calculated by ^1^H NMR according to the formula reported by Sen [59]:(1)$$Carbonyl\;content=\frac{I_{2}+I_{4}+I_{7}+I_{11}}{2\times \left(I_{2}+I_{4}+I_{7}+I_{11}\right)+\left(I_{5}+I_{8}+I_{12}\right)+(I_{9}+I_{13})}\times 100\%$$

I_2_, I_4_, I_5_, I_7_, I_8_, I_9_, I_11_, I_12_, and I_13_ are the integral of ^1^H NMR signal of corresponding methylene.

Extra ethylene (C2) insertion can be calculated by ^13^C NMR (signal of methylene) according to the formula reported by Chen [58]:(2)$$extra\;ethylene\;(C2)\;insertion=\frac{I_{4}+I_{7}+I_{11}}{I_{2}+I_{4}+I_{7}+I_{11}}\times 100\%$$

I_2_, I_4_, I_7_, and I_11_ are the integral of ^13^C NMR signal of corresponding methylene.

In the case of PE with in-chain ketones, in addition to carbonyl content, the proportion of different carbonyl groups (**1**, **3**, **6**, and **14**) in PKs also becomes important because the carbonyl content is too low to display obvious differences [51,52,60,61,62]. The ^13^C NMR signals of different carbonyl groups (**1**, **3**, **6**, and **14**) and adjacent methylene (**2**, **4**, **7**, and **15**) almost relatively isolate each other; therefore, the proportion of each kind of carbonyl can be calculated. Due to the low carbonyl content, the replacement of ^13^CO stock is used to enhance the sensitivity of the characteristic carbonyl [51,52]. Owing to the unique and intense signal of carbonyl, IR spectroscopy provides another practical measurement of the carbonyl content of PE with in-chain ketones, and a series of PKs with known carbonyl contents is need to draw a standard curve [52,63,64]. 

The relationship between the properties and microstructure of the copolymer has been systematically studied by Müller and Mecking [65,66]. Properties of PKs without a perfect alternating structure, such as *T*_m_, thermal decomposition temperature, and melt viscosity, for the microstructure, have a strong dependence on the copolymer, the CO content, and molecular weight, and the concentration of methyl branches along the chain directly determines the properties of the copolymer. In general, the properties of the copolymer are close to that of PE when the carbonyl content is low.

When the carbonyl content is high (>35%), the *T*_m_ of the copolymer decreases with the decrease of the carbonyl content because the multiple ethylene insertion destroys the dipolar interaction between the carbonyl groups in the alternating structure [50,67,68,69]. However, at a low carbonyl content (<30%), the properties of the copolymer are close to PE, and the decrease in carbonyl content usually leads to an increase in crystallinity, so the *T*_m_ of the copolymer decreases with the decrease in carbonyl content [59]. 

## 3. Catalytic Copolymerization Mechanism

Successive CO insertion is thermodynamically prohibited in CO/ethylene copolymerization [56]. Therefore, the key to obtain alternating or non-alternating copolymers is whether the ethylene sequential insertion with unfavorable kinetics can be achieved by the improvement of the catalyst system and the availability of copolymerization conditions.

Phosphine–sulfonate ligand palladium catalyst systems achieved the first example of non-alternating copolymerization of CO and ethylene, of which the catalytic mechanisms have been investigated scientifically [59,70,71,72]. Therefore, take the example of typical bidentate [P, P] palladium complexes (alternating copolymerization catalysts) and neutral phosphine–sulfonate ligated Pd complexes (Figure 1), the origins of sequential insertions of ethylene are proved to be (1) destabilization of Pd-acyl β-chelates **A** (phosphine-sulfonate); (2) facile decarbonylation of six-membered Pd-acyl chelates **D** [67,68,70]. As shown in Figure 1, five-membered Pd-acyl β-chelates **A** existing in CO/ethylene copolymerization catalyzed by bidentate [P, P] and phosphine–sulfonate palladium complexes, can accept insertions from ethylene or CO [67]. The stability of β-chelates **A** is lower in the case of phosphine–sulfonate ligands; therefore, the chelate cycle is easily opened by ethylene [68,71]. The binding affinities of CO and ethylene to β-chelates **A** have been measured by Sen in 2009 [59]. Although the ratio of binding affinities determined credibly by NMR seems reliable enough to explain the multiple insertions of ethylene, Sen believes decarbonylation also plays a crucial part in the non-alternation [59]. Thus, the second reason involving decarbonylation of six-membered Pd-acyl chelates **D** can be considered as a supplement to the first reason, which together constitute the reasons for the non-alternating copolymerization ability of neutral phosphine–sulfonate ligated palladium complexes. The relatively facile decarbonylation is also related to the instability of the corresponding Pd-acyl chelates **D** of phosphine–sulfonate ligands [59,71].

Although the destabilization of Pd-acyl β-chelates and decarbonylation could partly explain why non-alternating copolymerization occurs, most mechanisms cannot escape the scope of phenomenological theory. In fact, it is extremely difficult to find a direct connection between the ability of non-alternating copolymerization and the structure of ligands or catalysts. Therefore, there is still no perfect theory to predict new catalyst systems that can achieve non-alternating copolymerization of CO and ethylene.

However, some researchers have made outstanding contributions to exploring the nature of the origin of non-alternating copolymerization [72,73,74,75,76]. In different unique polymerizations realized by phosphine sulfonate palladium catalysts, similar *cis*/*trans* isomerization (Figure 2) of complexes and consistency in the high stability of cis *σ*-complex intermediates are useful to unravel the deep mechanism of non-alternating copolymerization [72,73]. 

*Cis*/*trans* isomerization is the connecting bridge of ethylene coordination and ethylene insertion. The *cis σ*-complex and *trans π*-complex are under equilibrium, and isomerization takes place readily in almost all cases. [72,73] Equilibrium with a high kinetic constant cannot directly determine the kinetically favorable step, because CO/ethylene non-alternating copolymerization is a result of kinetic regulation. However, the resulting concentration of the *trans π*-complex influences the kinetic trend of the monomer insertion to some extent [73]. In the case of CO/ethylene non-alternating copolymerization, sequential ethylene insertion must occur in the *trans π*-complex, and the concentration of the *trans π*-complex depends on the *cis*/*trans* equilibrium constant due to the stability of the *cis σ*-complex and *trans π*-complex and the structure of catalysts. More importantly, *cis*/*trans* isomerization is also assumed in recent novel examples of CO/ethylene non-alternating copolymerization [63,64], which implies that it might be a concordant insight into the non-alternating pathway.

Charge effects and orbital interactions were also proposed to explain the particularity of phosphine sulfonate ligands, and complexes with neutral charge and atoms with lone pairs covalent bonding Pd promote the multiple insertions of ethylene [2,73,74]. Orbital interactions specifically refer to an appropriate stereoelectronic effect of ligands and metal centers. The interactions involving lone pairs in *p* orbital on the ligated oxygen and *d_π_* orbital on palladium atom in the case of phosphine sulfonate ligands palladium catalyst systems (Figure 2). Generally, the SO_3_ group withdraws the *π*-electrons of the adjacent group. When an adjacent group connects directly with an oxygen atom, repulsion between one pair on the oxygen atom and *π*-electrons on Pd possibly causes an increase in the electron density of palladium. Resultantly, the back-donation from palladium to the antibonding orbital of ethylene is enhanced and the binding affinity of ethylene and metal may be strengthened, thereby enhancing the stabilization of the corresponding *trans π*-complex [53,74].

Notably, similar stereoelectronic structures exist in both diphosphazane monoxide and phosphinophenolate systems (Figure 3), which are two important catalysts for CO/ethylene non-alternating copolymerization (see Section 4). The striking similarity in stereoelectronic structure is the most critical factor in CO/ethylene non-alternating copolymerization. Orbital interactions provide a new approach to catalyst design and are helpful in developing new CO/ethylene non-alternating copolymerization catalysts.

## 4. Non-Alternating Copolymerization Catalysts

Due to the unfavorable kinetics of ethylene sequential insertion, designing catalysts for the synthesis of PKs without a perfect alternating structure is extremely difficult. Until recently, few examples of catalysts achieved ethylene sequential insertion. Phosphine–sulfonate Pd catalyst systems, diphosphazane monoxide (PNPO) Pd/Ni catalysts, and phosphinophenolate Ni catalysts represent the three main catalyst systems for CO/ethylene non-alternating copolymerization [50,51,58]. In addition, phosphanylferrocene-carboxylate ligand Pd catalysts [77] and zwitterionic Ni catalysts [78] were also reported to synthesize non-alternating PKs with low activity (<15 g PK·mmol^−1^·h^−1^).

### 4.1. Phosphine–Sulfonate Pd Catalysts

In 2002, Drent first reported the neutral palladium catalysts generated in situ from phosphine–sulfonate ligands [79]. The first example of a non-alternating linear CO/ethylene copolymer with successive ethylene incorporation was also reported by Drent by using a phosphine–sulfonate palladium catalyst system [50]. 

A new series of [P,O] palladium catalysts **Pd1**~**Pd3** (Figure 4) with *ortho*-alkoxy substituents generated in situ from Pd(OAc)_2_ and phosphine–sulfonate ligands were employed for CO/ethylene copolymerization. Despite the activities of copolymerization with these neutral catalysts (49~190 g PK·mmol·Pd^−1^·h^−1^) being low compared to classical 1,3-bis(diphenylphosphino)propane (dppp)-based palladium systems, these neutral catalysts produced non-alternating copolymers with 2.4~18.3% non-alt insertions (extra C2 insertion) [50]. The non-alternating insertions increase with the steric hindrance of the *ortho*-alkoxy substituents in the palladium catalytic system (O*^i^*Pr > OEt > OMe). Furthermore, the melting points of non-alternating copolymers made by catalysts from **Pd1** (234 °C), **Pd2** (231 °C) and **Pd3** (229 °C) reduce, reflecting the impact of steric hindrance in the palladium catalytic system on the microstructure of the resulting PKs.

The non-alternating CO/ethylene copolymerization using phosphine–sulfonate palladium systems is attributed to the fact that ethylene insertion can take place from the most stable chelate complex “*cis σ*-complex” via the intermediate “*trans π*-complex” (see the “Catalytic copolymerization mechanism” section) [73]. By examining the kinetic and thermodynamic parameters, Sen discovered that the catalyst exhibits a remarkably minor disparity in its binding affinities for ethylene and carbon monoxide [59]. However, the difference in monomer binding affinity alone is not sufficient to explain the degree of non-alternation actually observed in this system. Thus, decarbonylation plays a significant role in the non-alternation. It has been observed that the steric bulk prevents the formation of chelate bonds between the carbonyl oxygens of the PKs chain and the metal center, which facilitates the decarbonylation of an acyl group. This process leads to successive ethylene incorporation and ultimately results in the formation of non-alternating units [59,70,71]. 

Although successive CO insertions are thermodynamically impossible, CO exhibits a stronger binding affinity to the active palladium species and tends to insert more readily into the Pd-alkyl bond than ethylene. Consequently, obtaining PKs with low CO content is greatly challenging. Sen and coworkers successfully achieved the formation of non-perfect alternating PKs with as low as 10 mol% CO in the **Pd1** (Figure 4) catalytic system, although the yields were relatively low [69]. Sen and coworkers systematically controlled the monomer feed ratio and reaction temperature, and non-perfect alternating PKs with CO incorporation ranging from 0 to 50 mol% were produced [59].

Muller successfully prepared low-molecular-weight non-alternating PKs (<6 kg·mol^−1^) with varying CO contents (0~50 mol%) in the **Pd1** (Figure 4) [65]. Segments with high CO density result in domains with strong intermolecular dipole–dipole interactions, which play a dominant role in the phase change behavior of PKs. Additionally, the melt viscosity decreases rapidly with decreasing molecular weight of PKs. The crystallinity of the non-perfect alternating PKs is directly related to the -(C_2_H_4_)_n_- segment length. Additionally, the presence of methyl branches along the chain significantly affects the material properties in the solid state, resulting in reduced crystallinity and lower enthalpy of melting.

The unique properties of phosphine–sulfonate palladium catalysts have sparked widespread research interest in the synthesis of non-alternating PKs. Rieger and coworkers reported the palladium catalysts **Pd4**~**Pd6** (Figure 4) with modification of both the benzene ring and phosphine moieties for non-alternating CO/ethylene copolymerizations [67]. **Pd4** does not produce any significant amounts of polymer under the conditions used, and **Pd5** and **Pd6** can yield high molecular weight PKs (*M*_w_ up to 370 kg·mol^−1^) with 2.4~18.3% extra C2 insertions and a melting temperature of 220~230 °C.

Rieger further reported the palladium catalysts **Pd7**~**Pd9** (Figure 5) and two monosubstituted palladium catalysts, **Pd10** and **Pd11** (Figure 5), for CO/ethylene copolymerizations [67]. Additionally, **Pd7** does not produce PK, but **Pd8** and **Pd9** exhibit high polymerization activities (2.5~106 g PK·mmol·Pd^−1^·h^−1^) and thermal stability compared to the corresponding in situ generated palladium catalysts. **Pd9** also exhibits substantial copolymerization activity of 75 g PK·mmol·Pd^−1^·h^−1^ at 200 °C. However, the obtained PK has lower extra C2 insertion (<4.9 mol%) [67]. When 2 wt% B(C_6_F_5_)_3_ was used as an activator, the monosubstituted palladium **Pd10** and **Pd11** (Figure 5) showed increased copolymerization activities (up to 292 g PK·mmol·Pd^−1^·h^−1^) to produce PKs with remarkably high extra ethylene insertions (up to 26.4%) [67].

In addition to modifying the phosphine moiety, Bianchini and Oberhauser conducted a comparative study on the impact of the backbones of phosphine–sulfonate ligands on the performance of palladium catalysts [68]. They investigated the non-alternating CO/ethylene copolymerization catalyzed by palladium catalysts **Pd12**~**Pd15** (Figure 6) with modified phosphine–sulfonate ligands. **Pd12** and **Pd14** with rigid phenyl backbones exhibited much higher activity (41~607 g PK·mmol·Pd^−1^·h^−1^) than **Pd13** and **Pd15** (0.4~5.4 g PK·mmol·Pd^−1^·h^−1^) with the more flexible ligands. The presence of benzoquinone (BQ) oxidant has no appreciable influence on extra C2 insertion while increasing productivity in copolymerization catalyzed by **Pd12**. Additionally, the former produces copolymers with a significantly higher molecular weight (*M*_w_ > 30 kg·mol^−1^). The authors think that the decreased copolymerization activities observed in **Pd13** and **Pd15** with flexible ligands could be mainly attributed to the faster degradation of catalytically active species. Furthermore, palladium catalysts **Pd12** and **Pd14** with a rigid backbone produced PKs with higher extra-ethylene insertions (up to 27.8 mol%) compared to **Pd13** and **Pd15** (0.2~23.2 mol%).

The ferrocene moiety provides quite a variety of possibilities for electronically modifying catalyst systems derived from chelate ligands. Erker and coworkers reported two in situ phosphine–sulfonate palladium catalysts, **Pd16** and **Pd17** (Figure 7), and **Pd18** complexes with a ferrocene-derived ligand for CO/ethylene copolymerization [80]. At a relatively low CO/ethylene feed, these catalysts produced nearly alternating PKs. When the CO/ethylene partial pressure ratio is approximately 1:10, substantially non-alternating PKs were obtained. The overall catalyst activities decreased as the CO partial pressure decreased, but the activities were still quite remarkable (18~159 g PK·mmol·Pd^−1^·h^−1^). **Pd17** exhibited higher copolymerization activity than **Pd16** in spite of a lower percentage of extra ethylene insertions (<1 mol%). The addition of B(C_6_F_5_)_3_ as a Lewis acid activator increased the activity of **Pd18** while decreased extra C2 insertion. By precisely controlling the conditions, it is possible to prepare PKs with different amounts of extra ethylene (1~27 mol%) and varying polymer melting temperatures (167~242 °C).

Nozaki also reported phosphine–sulfonate palladium catalysts **Pd19**~**Pd22** (Figure 8) for the copolymerization of ethylene with metal carbonyls as the CO source [60]. In contrast to conventional CO/ethylene copolymerization reactions, excellent non-alternating selectivity has been achieved with moderate activity (1.2~54 g PK·mmol·Pd^−1^·h^−1^). Linear high molecular weight (44 kg·mol^−1^) polyethylene bearing low content isolated in-chain carbonyls (<4%) was obtained. Although the copolymer retained the properties of polyethylene, it exhibited faster degradation under UV light irradiation compared to pure PE.

Jian reported a non-alternating (>99%) terpolymerization of CO with ethylene and fundamental polar monomers using palladium catalysts **Pd23**~**Pd26** (Figure 8) [81]. This palladium system effectively prevented the formation of undesired polar monomer chelates such as acrylate (MA, BA), acrylic acid (AA), vinyl ether (VE), and acrylonitrile (AN). High molecular weight linear PE was synthesized with both in-chain isolated carbonyl (>99%) and main-chain functional groups (0.18~2.38 mol%). The incorporation of these low-content isolated carbonyls allows for the photodegradation of the PE. Additionally, the introduction of polar functional groups significantly enhances their surface properties.

Table 3 lists a summary of representative results for non-alternating copolymerization activity, extra C2 insertion, and melting temperature in the phosphine–sulfonate palladium catalyst systems.

### 4.2. Diphosphazane Monoxide Pd/Ni Catalysts

In 2018, Chen discovered a new type of electronic asymmetric diphosphazane monoxide (PNPO) ligand (Figure 5) with multiple modifiable moieties for copolymerization of ethylene with polar monomers [82,83,84]. The short-bite ligand platform increases ligand rigidity, and unique electronic asymmetry achieved by a strongly *σ*-donating phosphine moiety and a weakly *σ*-donating phosphine oxide moiety is believed to be responsible for its unexpectedly excellent catalytic property. PNPO Ni/Pd catalysts for CO/ethylene non-alternating copolymerization are shown in Figure 9, and the corresponding non-alternating copolymerization results are summarized in Table 4.

Chen developed a series of catalysts **Pd27**~**Pd40** (Figure 9) with different electronic effect substituents on phosphine moieties (Ar^1^, Ar^2^) and N-aryl moieties (Ar^3^) [58]. Notably, the introduction of electron-donating substituents at the N-aryl moiety (Figure 10) of PNPO ligands (**Pd27**~**Pd28**, **Pd29**~**Pd32**, **Pd33**~**Pd35**, and **Pd36**~**Pd37**) caused increasing activities and *M*_n_. More importantly, the carbonyl content also decreased as the electron-donating ability (–CF_3_ < –H < –*^t^*Bu < –OMe < –NMe_2_) of the substituents at the N-aryl moiety increased [57]. The introduction of different substituents on phosphine moiety (**Pd32**, **Pd35**, **Pd37**, **Pd38**, and **Pd39**) significantly affects the activity and carbonyl content in the copolymer. **Pd38** with Ar_3_ = 2,6-OMeC_6_H_3_ exhibits the highest activity and the lowest carbonyl content, while **Pd39** with stronger electron-donating Ar_3_ = 2-NMe_2_C_6_H_4_ is inactive because of the coordination mode change [57]. Catalyst **Pd40** bearing isopropyl affords PK with high carbonyl content (49%) and low activity (25.5 g PK·mmol·Pd^−1^·h^−1^). The representative catalyst **Pd38** is highly efficient for CO/ethylene non-alternating copolymerization. With an increasing CO/ethylene feed ratio from 1/1 to 1/20, the resultant PKs have alternating to non-alternating structures with a decreasing melting temperature (*T*_m_) from 248 °C to 147 °C [58].

Liu has reported **Pd41**~**Pd50** for non-alternating copolymerization of CO and ethylene [63]. Compared with **Pd32**, **Pd41**~**Pd43** bearing an aliphatic substituent on the amine moiety showed non-alternating copolymerization with almost two times higher activity and higher carbonyl content. **Pd44** bearing benzylamine and Ar_3_(2,6-OMeC_6_H_3_) (Figure 10) is highly efficient (activity up to 124.1 g PK·mmol·Pd^−1^·h^−1^) for CO/ethylene non-alternating copolymerization. Liu also found that steric hindrance and electron-donating ability of the substituent on phosphine oxide moiety are beneficial to extra ethylene insertion, thus **Pd45**~**Pd49** afforded PKs with relatively high extra ethylene insertion (12.2%) [63]. In addition, **Pd50** bearing -NMe_2_ on the phosphine oxide moiety has a different complexation mode, and it afforded PKs with low activity and carbonyl content.

Liu also developed a series of PNPO nickel/palladium catalysts, and PE with in-chain ketones was synthesized by **Ni1**~**Ni8**, **Pd29**~**Pd32**, and **Pd38** [61]. In a very low CO/ethylene feed of 0.2%, copolymers were produced in low carbonyl content and low *T*_m_ (<140 °C). Both steric hindrance and the electronic nature of ligands have significant impacts on activity, *M*_n_, and carbonyl content. With increasing electron-donating ability of the substituents at the N-aryl moiety from **Ni1** to **Ni4**, the resultant copolymers’ *M*_n_ decreases, but their carbonyl content and activity exhibited uncertain regularity. **Ni5** and **Ni8** bearing the strongest electron-donating and electron-withdrawing substituents on phosphine moiety are inactive, while **Ni6** and **Ni7** afforded a copolymer with the highest activity (301 g PK·mmol·Ni^−1^·h^−1^). Palladium-based catalysts **Pd29**~**Pd32** and **Pd38** were also used to prepare PE with in-chain ketones. The electron-donating substituent on arylamine moieties (**Pd29**~**Pd32**) increased both activity and *M*_n_ [58,61]. **Pd38** bearing a strong electron-donating phosphine moiety is the most efficient and afforded copolymers with the highest *M*_n_ (93.3 kg·mol^−1^) in 0.1% CO/ethylene feed.

### 4.3. Phosphinophenolate Ni Catalysts

Copolymerization of CO and ethylene catalyzed by Ni complexes is extremely challenging owing to the strong electrophilic nature of Ni. Most Ni catalysts only produced alternating PKs [85,86,87,88,89]. Phosphinophenolate nickel catalysts were first developed for the copolymerization of ethylene and acrylates [90,91]. In 2021, Mecking reported CO/ethylene non-alternating copolymerization using phosphinophenolate Ni complexes (**Ni9**~**Ni14**) to yield in-chain ketone PE materials. In 0.2 mol % CO feed, PE with in-chain ketones was obtained with a *T*_m_ low of 134 °C [51]. This material maintains the excellent mechanical and processing properties of traditional PE and also possesses the degradability of PKs [51]. Phosphinophenolate Ni catalysts for CO/ethylene non-alternating copolymerization are summarized in Figure 11, and the corresponding non-alternating copolymerization results are summarized in Table 5.

Generally, phosphinophenolate Ni complexes have a relatively complex structure and multiple modification modes [51,64]. Unlike phosphine sulfonate Pd catalysts, the rate-determining step of alternating and non-alternating steps in phosphinophenolate Ni-catalyzed CO/ethylene copolymerization were identified as the opening of the six-membered *C,O*-chelate by ethylene coordination and *cis*/*trans* isomerization of an alkyl-olefin intermediate, respectively (Figure 12) [64]. Short duration copolymerization experiments and theoretical studies were systematically studied, and the details of the connection between the catalyst structure and rate-determining step were unraveled. The steric effect at the aromatic rings on the phosphine moiety apically coordinated to the metal center significantly impacts the free energy barrier of *cis*/*trans* isomerization and the free energy barrier of the ethylene coordination step (Figure 12) [64]. Simultaneously, an *η*^2^-coordination of a P-bound aromatic moiety axially oriented to the metal center, which is considered as a crucial feature of catalysts, modulates the competition between alternating and non-alternating pathways.

In general, the influence of catalyst structure on competition can be summarized as follows. (1) Bulkier steric hindrance, enhanced by the *η*^2^-coordination mentioned above, impedes the inversion of the intermediate conformation during copolymerization, thereby increasing the energy barrier of *cis*/*trans* isomerization and finally weakening the tendency for non-alternating polymerization. (2) The electronic effect of ligands directly determines the electron density at the Ni center. The strong electrophilicity of Ni easily causes a very stable resting state, which prohibits both alternating and non-alternating pathways. In contrast, strong electron-donating groups lead to low electrophilicity Ni, which is favorable for alternating polymerization because of the easy opening of the six-membered *C,O*-chelate [51,64]. Interestingly, phenolate substitution has few effects on the microstructure of copolymers compared with phosphine substituents [64].

In addition, Nozaki also reported phosphinophenolate Ni catalysts for the synthesis of PE with in-chain ketones using metal carbonyls as a carbonyl source [62]. High selectivity to isolated carbonyl and higher activity were realized compared to the Pd catalysts reported by them.

## 5. Conclusions and Outlook

Aliphatic PKs generated from the copolymerization of CO and ethylene have attracted considerable attention over the last decades due to their unique properties and environmental friendliness. Alternating PKs have a high *T*_m_ and insolubility, while PKs without a perfect alternating structure have better processability. However, the synthesis of PKs without a perfect alternating structure is extremely difficult because of the unfavorable kinetics of ethylene sequential insertion. Phosphine–sulfonate Pd catalyst systems, PNPO Pd/Ni catalysts, and phosphinophenolate Ni catalysts represent the three main catalysts for CO/ethylene non-alternating copolymerization. Catalytic activity and the carbonyl content of PKs are two important parameters to evaluate catalysts. Interestingly, stereoelectronic structure and the orbital interactions involving lone pairs in the *p* orbital on the oxygen donor and *d_π_* orbital on the metal atom are observed in three kinds of catalysts, which open a window for the design of non-alternating ethylene/CO copolymerization catalysts.

Despite significant breakthroughs in non-alternating copolymerization of CO in recent years, there are still several challenges for future development in this field: (1) modulating the carbonyl content of PKs by CO/ethylene feed in copolymerization often causes a rapid decrease in activity. (2) There is no widely acceptable theory to explain the relationship between the structure of catalysts and non-alternating copolymerization, therefore designing new catalysts seems to only depend on a large number of experiments. (3) Noble palladium catalyst represents the most successful catalytic system, while the replacement of palladium by nickel or other non-noble transition metals poses significant challenges.

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
