# Peer review of "Recent Advances in Synthesis of Non-Alternating Polyketone Generated by Copolymerization of Carbon Monoxide and Ethylene"

_ijms, 2024, doi:10.3390/ijms25021348_

Round 1

Reviewer 1 Report

Comments and Suggestions for Authors

Thank you for sending this review article entitled “Recent advances in synthesis of non-alternating polyketone generated by copolymerization of carbon monoxide and ethylene”. [ijms-2824596]

In this review article, the authors summarize the recent research activities in the metal-catalyzed synthesis of aliphatic polyketones, especially the non-perfect alternating copolymers that show better processability. The authors present an unbiased summary of the various types of the used Pd and Ni catalysts with different ligands. The mechanisms for the synthesis of alternating and non-perfect alternating PKs were also demonstrated along with clarifying the crucial role of the catalyst system and copolymerization conditions. .

The review article was written as a whole in a good way with proper flow. The authors shed light on current challenges are need to be overcome in the near future. 

I recommend this paper for publication after minor revision which should be corrected before publishing this manuscript.

- In Table 3, the authors did not explain the role of B(C6F5)3 as phosphine scavengers, or using BQ as an organic oxidant (also you need to show that BQ: benzoquinone).

- Table 3 on page 11 line 333 needs to be revised to Table 4

- Table 4 on page 14 line 409 needs to be revised to Table 5

- There are some tiny typographic mistakes you need to revise carefully

For example, page 8 line 278 wtih should be with

- The References section needs to be revised carefully to consistent

For example, Refs 6, 69, 10, 15, etc.

Author Response

  1. In Table 3, the authors did not explain the role of B(C6F5)3 as phosphine scavengers, or using BQ as an organic oxidant (also you need to show that BQ: benzoquinone).

Response: B(C6F5)3 was called as “activator” and “Lewis acid activator” in the references 67 and 88. In the table footnote, we supply B(C6F5)3 is an activator and the full name of BQ.

Besides, we also supply the effect of B(C6F5)3 activator and BQ oxidant in the text.

  1. Some errors

Table 3 on page 11 line 333 needs to be revised to Table 4

Table 4 on page 14 line 409 needs to be revised to Table 5

There are some tiny typographic mistakes you need to revise carefully

For example, page 8 line 278 wtih should be with

The references section needs to be revised carefully to consistent

For example, Refs 6, 69, 10, 15, etc

Response: Thanks for reviewer’s kindness and carefulness. There errors have been revised carefully. All references were revised carefully to consistent.

Reviewer 2 Report

Comments and Suggestions for Authors

It is a comprehensive well written review on recent advances in synthesis of non-alternating polyketone generated by copolymerization of carbon monoxide and ethylene. In the catalytic synthesis of this kind of polymeric materials, please add a short paragraph on the organization and illustration of the most representative academic research papers on each important topic in a graphic way. In addition, the relevant application examples of these engineering plastics are also briefly explained.

Author Response

It is a comprehensive well written review on recent advances in synthesis of non-alternating polyketone generated by copolymerization of carbon monoxide and ethylene. In the catalytic synthesis of this kind of polymeric materials, please add a short paragraph on the organization and illustration of the most representative academic research papers on each important topic in a graphic way. In addition, the relevant application examples of these engineering plastics are also briefly explained.

Response: We have revised Figure 1 accordingly. Polymer structure, application examples of these engineering plastics, representative academic research papers are clearly shown. In the text, we also decriable the relevant application of these three polymeric materials in brief.

Besides, we revise some typographic mistakes and highlight in manuscript.
